# Evaluation of Portal Vein Doppler Findings in Patients with Crimean–Congo Hemorrhagic Fever

**DOI:** 10.3390/medicina55050146

**Published:** 2019-05-16

**Authors:** Erdal Karavaş, Faruk Karakeçili

**Affiliations:** 1Department of Radiology, Faculty of Medicine, Erzincan Binali Yıldırım University, 24100 Erzincan, Turkey; 2Departments of Infectious Diseases and Clinical Microbiology, Faculty of Medicine, Erzincan Binali Yıldırım University, 24100 Erzincan, Turkey; drfarukkarakecili@hotmail.com

**Keywords:** Crimean–Congo hemorrhagic fever, ultrasonography, doppler, portal vein, disease severity

## Abstract

*Background and objectives*: In this study, we compare portal vein Doppler ultrasound (US) findings between patients with Crimean–Congo hemorrhagic fever (CCHF) and healthy persons and investigate the practicability of these findings in the prediction of disease severity. *Materials and Methods:* In this prospective study, portal vein Doppler US was performed in patients diagnosed with CCHF and healthy persons between March 2016 and May 2018. The patients were grouped according to mild-to-moderate and severe progression of CCHF. Liver size, portal vein diameter, portal vein flow rate, spleen volume, and splenic vein diameter were recorded in the patients and healthy controls. *Results*: Of the 48 patients diagnosed with CCHF, 25 were male. According to the scoring made, 38 patients were evaluated as having mild-to-moderate disease progression, and 10 were evaluated as having severe disease progression. With respect to the Doppler US findings, liver size, spleen volume, portal vein diameter, splenic vein diameter, and portal vein flow rate were significantly higher in the patient group compared with the controls. However, no significant difference was found in these parameters between the severe and mild-to-moderate progression groups. *Conclusions*: In the evaluation of and follow-up with patients with CCHF, portal vein Doppler US is a non-invasive and reliable tool for diagnosis.

## 1. Introduction

Crimean–Congo hemorrhagic fever (CCHF) is a zoonotic disease characterized by a hemorrhage and fever [1,2]. It is caused by Crimean–Congo hemorrhagic fever virus (CCHFV), which belongs to the *Orthonairovirus* genus of the *Nairoviridae* family [3]. The virus is carried by *Hyalomma spp*. ticks, particularly *Hyalomma marginatum marginatum* [2,4,5]. Transmission to humans occurs via a bite from a tick infected with the virus or through contact with tissue or blood of viremic animals [6,7].

CCHFV infection is seen in Asia, Europe, Africa, and the Middle East [8]. The disease commonly affects individuals living in rural areas and those involved in husbandry [9,10]. The first case of CCHF in Turkey was identified in 2002 [10]. Up to 2016, the disease had affected 10,219 individuals and resulted in the deaths of 485 individuals [11].

The clinical disease spectrum of CCHFV infection varies from mild-to-moderate to severe. Several scoring systems have been developed to determine the severity of the disease based on the clinical and laboratory follow-up. These scoring systems are used to predict the course and mortality of the disease [12,13].

In this study, for the first time in the literature, we performed portal vein Doppler ultrasound (US) screening in CCHF patients and compared the outcomes with those of healthy individuals. The aim of the study was to investigate the practicability of using radiological findings to predict the severity of CCHF in the early period in patients with mild-to-moderate and severe progression.

## 2. Materials and Methods

### 2.1. Patients

Before beginning the study, approval was received from the ethics committee of Erzincan Binali Yildirim University (9/5/2016). The study was then conducted prospectively after obtaining informed consent from the patients. In total, 49 CCHF patients hospitalized and treated in our hospital between March 2016 and June 2018 and 51 healthy volunteers were included in the study. One patient was excluded from the study because of an inability to hold their breath. Thus, 48 patients completed the study. All the patients were called to a follow-up visit 1 month after being discharged following recovery. Only 17 of the recovered patients came to the follow-up visit, and Doppler US examinations were performed.

The patients’ demographic characteristics, such as age, sex, epidemiological history, occupation, date of admission, existing symptoms, physical examination findings, and daily laboratory outcomes were recorded. Based on their scores [12,13], the patients were grouped based on those with mild-to-moderate disease progression and severe disease progression.

### 2.2. Laboratory Tests

Laboratory tests were closely monitored at the time of admission and during daily follow-up examinations. Accordingly, hemoglobin, platelet count, white blood cells (Sysmex XN-2000, Sysmex Corporation, Kobe, Japan), aspartate aminotransferase, alanine aminotransferase, creatinine phosphokinase, lactate dehydrogenase, and international normalized rate (INR) (Beckman Coulter AU2700, Beckman Coulter, Brea, CA, USA) were recorded. The diagnosis of CCHF was confirmed by the Turkish Public Health Institution, Microbiology Reference Laboratories Department, National Virology Reference Central Laboratory. The diagnosis was established using real-time reverse transcriptase polymerase chain reaction (RT-PCR) and immunofluorescence assay methods. Viral RNA was isolated using a high pure viral nucleic acid kit (Roche Diagnostics GmbH, Mannheim, Germany). The presence of CCHFV RNA was tested using TaqMan probe-based single-reaction RT-PCR as described by Yapar et al. [14]. CCHF viral immunoglobulin M antibodies were detected using an immunofluorescence assay in accordance with the instructions of the manufacturer (CCHFV Mosaic 2; Euroimmun, Lübeck, Germany).

### 2.3. Doppler US Evaluation

Following the diagnosis of CCHF, color Doppler US of the portal vein was performed by a radiologist with 9 years of experience using a 2.5-MHz convex transducer (Hitachi Hi Vision Preirus; Tokyo; Japan). The procedure was performed between 1 and 10 days (mean 3.6 ± 2.6) after the onset of symptoms. Repeat Doppler screening of 17 patients was conducted at a follow-up visit 1 month after being discharged following a full recovery.

The Doppler US screening procedure was performed according to standard protocols, and the liver and spleen were examined using gray-scale B-mode. During the examination, the size of the liver was measured from the craniocaudal length to the midclavicular line, with the patients in a supine position, and the portal venous diameter was measured at the level of the hilus. Splenic width (W), thickness (T), maximum length (ML), and craniocaudal length (CCL) were measured. The splenic volume was calculated using the following formula [15]: 0.524 × W × T × (ML + CCL)/2.

The splenic venous diameter was measured from the level of the hilus. The portal venous flow rate was measured from the level of the portal vein hilus, with the angle adjusted to 60° so as to be parallel to the vascular structure under 60°.

### 2.4. Statistical Analysis

The results were expressed as mean ± standard deviation and median (min–max) for continuous variables and as “*n*” and percentages (%) for categorical variables. The normality of the variables was tested with the Kolmogorov–Smirnov test when examining statistically significant differences between the groups in terms of continuous variables. Power analysis was performed with G*power version 3.1.9.2. Data with normal distribution were compared using the Student’s t test otherwise Mann–Whitney U was used. Pearson’s chi-square test was used in the analysis of categorical variables. A value of *p* < 0.05 was considered statistically significant. Data were analyzed using IBM SPSS version 19 (IBM, Armonk, NY, USA).

## 3. Results

Of the 48 patients, 25 were males. The median age was 50.3 (range: 19–79) years. There was no statistically significant difference between the patients and controls in terms of age and sex (*p* > 0.05). In the study group, 38 (79.2%) of the 48 patients were involved in farming/husbandry, and the other 10 (20.8%) patients belonged to other professions.

The most common symptoms of the patients included weakness, a fever, diffuse body pain, headache, nausea/vomiting, and abdominal pain. In the physical examination, tachycardia was found in 4 (8.3%) patients, and a mental fog was recorded in 1 (2.1%) patient. The laboratory findings at the time of admission revealed thrombocytopenia in 46 (95.8%) patients, leukopenia in 42 (87.5%) patients, and anemia in 3 (6.3%) patients. Aspartate aminotransaminase and alanine aminotransaminase were elevated in 38 (79.2%) patients, lactate dehydrogenase was elevated in 31 (64.6%) patients, creatine kinase was elevated in 34 (70.8%) patients, and the international normalized ratio was increased in 5 (10.4%) patients. The B-mode examination revealed hepatomegaly in 8 (16.7%) patients, splenomegaly in 13 (27.1%) patients, increased portal venous diameters in 18 (37.5%) patients, and increased splenic venous diameters in 4 (8.3%) patients. The portal venous flow rate was raised in 19 (39.6%) patients (Table 1). Abdominal free fluid was detected in only 1 (2.1%) patient, and no thickening of the gallbladder wall was found in any patient.

According to the scoring of disease severity recommended by Bakir et al. [12,13], 10 patients had severe disease progression, and the other 38 patients had mild-to-moderate disease progression. Following treatment, all the patients recovered and were discharged from the hospital.

The liver size, splenic volume, and portal venous and splenic venous diameters in the patient group were significantly higher than those in the controls (*p* < 0.001). In the Doppler examination, the portal venous flow rates of the patients were significantly higher than those of the controls (*p* = 0.002) (Table 2).

There was no statistically significant difference between the severe and mild-to-moderate disease progression groups in terms of liver size, splenic volume, portal venous and splenic venous diameters, and portal venous flow rates (*p* > 0.05) (Table 3).

In the Doppler US examination of the 17 patients after recovery, there were significant improvements in the liver sizes, splenic volumes, and portal venous and splenic venous diameters (*p* < 0.05). However, there was no significant improvement in portal venous flow rates (*p* > 0.05) (Table 4).

## 4. Discussion

There are limited studies in the literature on radiological findings of patients with CCHF. Our study is the first in the literature to investigate the portal vein with Doppler US. In addition, it is the first study to evaluate the radiological findings of the patients after recovery. In our study, the liver size, splenic volume, and portal venous and splenic diameters were significantly higher in the patient group as compared with those in the controls.

Since 2002, there has been a concerning increase in the number of CCHF reports in Turkey [10]. Those affected are predominantly of active working age and involved in farming/husbandry. Healthcare workers, slaughterhouse workers, and farmworkers are commonly affected by CCHFV infection. The female-to-male ratio of CCHF cases in Turkey is almost equal due to the number of women involved in agricultural-related work [16,17,18]. In the present study, 79.2% of the patients had a farming/husbandry background. The number of male patients (25/48) slightly exceeded that of female patients.

CCHF symptoms usually have an acute onset, with fever and weakness as the first symptoms, followed by a headache, fatigue, and the development of diffuse muscular and articular pain. In cases of severe progression, the patient may become agitated and have a mental fog [6,7,19]. A tendency toward bleeding is also found in CCHF patients, with subcutaneous hemorrhages, nasal bleeding, gingival bleeding, hematuria, hematemesis, melena, and bleeding of the visceral organs reported in previous studies. Primary laboratory findings include thrombocytopenia, leucopenia, elevated liver enzyme levels, and prolonged bleeding markers. CCHF is a potentially fatal disease, which may have a severe progression, with multiple organ involvement [19,20]. In our study, the physical examination findings included fevers, headaches, weakness, and diffuse muscular and articular pain. Thrombocytopenia (95.8%) and leukopenia (87.5%) were the most common laboratory findings, consistent with the literature [11,17,21]. All the patients included in our study recovered, and no mortality occurred following discharge.

Previous studies investigated the relationship between radiological findings and disease severity in CCHF. Using abdominal US, Ziraman et al. reported hepatomegaly (25%), splenomegaly (19%), and abdominal free fluid (11%) in CCHF patients [22]. In the same study, the incidence of hepatomegaly, a thickened gallbladder wall, and the presence of abdominal free fluid were statistically significantly higher in patients with severe disease progression as compared with those with mild-to-moderate disease progression. In another study on CCHF patients, Ozmen et al. reported hepatomegaly (56.9%), splenomegaly (19.6%), and abdominal free fluid (70.6%) using abdominal computed tomography (CT) [23]. In the same study, the laboratory findings and CT outcomes were compared, and the presence of abdominal free fluid was statistically significant in patients with a platelet count lower than 50,000. As compared with the reports in the literature [22,23], the incidence of hepatomegaly was lower (16.7%) in the present study, and the incidence of splenomegaly was higher (27.1%). In the present study, abdominal free fluid was found in only one patient. Thus, the incidence was much lower (2.1%) than that reported in the earlier studies [22,23]. In the study by Ziraman et al., the radiological investigations were performed a mean of 7.8 days after the onset of symptoms, whereas they were performed a mean of 3.6 days after symptom onset in our study. We think that the lower incidence of abdominal free fluid in the present study may be due to the early timing of the radiological examinations. The latter may also explain the absence of a thickened gallbladder wall in all the patients in our study. In the present study, there was no statistically significant difference between the patients with severe disease progression and those with mild-to-moderate disease progression in terms of the studied parameters.

Two previous studies investigating Doppler findings in CCHF patients evaluated carotid and vertebral artery Doppler findings [11,24]. Salk et al. found that peak end systolic and diastolic flow rates of bilateral carotid arteries, internal carotid arteries, and vertebral arteries, in addition to resistive and pulsatility indices, were elevated in CCHF patients [24]. Similar to the study by Salk et al. [24], Karavas and Karakecili detected increases in peak end systolic and diastolic flow rates of vertebral arteries in patients with CCHF [11]. In the same study, the total cerebral flow rate was high [11]. These results point to an increase in cerebral hemodynamics in CCHF patients. In our study, the portal venous flow rate in the patient group was significantly higher as compared with that in the controls.

In the current study, for the first time in the literature, repeat Doppler screening of CCHF patients (17/48) was performed 1 month after discharge, following a full recovery. The results of the Doppler examination showed a significant improvement in liver sizes, splenic volumes, and portal venous and splenic venous diameters but no significant improvement in portal venous flow rates. This might be due to the small number of patients in the control sample. It is also possible that the portal venous flow rate takes longer than the other studied parameters to return to normal.

The most important limitation of our study is the small patient population and the low number of severe cases. Therefore, the assessment of disease severity in the patient population was not optimal.

## 5. Conclusions

In conclusion, CCHFV remains a seasonal problem worldwide, including in Turkey. Portal vein Doppler US is a non-invasive and reliable diagnostic tool for the evaluation of and follow-up with patients with CCHFV infection and should be considered in the management of such cases.

## Figures and Tables

**Table 1 medicina-55-00146-t001:** Demographic characteristics, symptoms, physical examination laboratory, and Doppler ultrasound (US) findings of the patients.

Variable	Disease Severity				Total	
	Mild-to-Moderate		Severe			
	*n*	%	*n*	%	*n*	%
**Sex**						
Male	20	52.6	5	50	25	52.1
Female	18	47.4	5	50	23	47.9
**Profession**						
Farming/husbandry	37	97.4	8	80	38	79.2
Other	1	2.6	2	20	10	20.8
**Most common symptoms**						
Weakness	37	97.4	10	100	47	97.9
Headache	35	92.1	8	80	43	89.6
Diffuse body pain	34	89.5	9	90	43	89.6
Fever (37.5°)	33	86.8	8	80	41	85.4
Nausea/vomiting	20	52.6	5	50	25	52.1
Diarrhea	10	26.3	4	40	14	29.2
**Physical finding**						
Tachycardia	3	7.9	1	10	4	8.3
Mental fog	1	2.6	0	0	1	2.1
Hematuria	1	2.6	0	0	1	2.1
Gastrointestinal system hemorrhage	0	0	1	10	1	2.1
**Laboratory features**						
Thrombocytopenia (<150,000/mm^3^)	36	94.7	10	100	46	95.8
Leucopenia (<4000/mm^3^)	34	89.5	8	80	42	87.5
Elevated AST/ALT (>40 U/L)	32	84.2	6	60	38	79.2
Elevated CK (>240 U/L)	27	71.1	10	100	34	70.8
Elevated LDH (>450 U/L)	23	60.5	8	80	31	64.6
Elevated INR (>1.2 Sn%)	3	7.9	2	20	5	10.4
Anemia (<12.5 gr/dL)	1	2.6	2	20	3	6.3
**Ultrasound finding**						
Hepatomegaly (>160 mm)	5	13.2	3	30	8	16.7
Splenomegaly (>120 mm)	9	23.7	4	40	13	27.1
Elevated portal vein diameter (>10 mm)	13	34.2	5	50	18	37.5
Elevated splenic vein diameter (>13 mm)	3	7.9	1	10	4	8.3
Elevated portal vein flow rate (>40 cm/s)	15	39.5	4	40	19	39.6
Abdominal free fluid	1	2.6	0	0	1	2.1

AST, aspartate aminotransferase; ALT, alanine aminotransferase; CK, creatine kinase; LDH, lactatedehydrogenase; INR, International normalized ratio.

**Table 2 medicina-55-00146-t002:** Liver size, spleen volume, portal vein and splenic vein diameters, and portal vein flow rates in the patient and control groups.

Parameter	CCHF		Normal		*p* Value
	*n*	Values Mean ± SD or Median (Min–Max)	*n*	Values Mean ± SD or Median (Min–Max)	
Liver size (mm)	48	147.30 ± 14.89	51	136.76 ± 11.70	<0.001
Splenic volume (cm^3^)	48	294.71 ± 107.19	51	175.96 ± 78.76	<0.001
PV diameter (mm)	48	12.36 ± 1.74	51	9.50 ± 1.60	0.002
SV diameter (mm)	48	6.65 (3.00–10.90)	51	3.90 (1.70–8.70)	<0.001
PV velocity (cm/s)	48	27.92 ± 7.47	51	23.77 ± 4.73	<0.001

CCHF, Crimean–Congo hemorrhagic fever; PV, portal vein; SV, splenic vein; SD, standard deviation.

**Table 3 medicina-55-00146-t003:** Liver size, spleen volume, portal vein and splenic vein diameters, and portal vein flow rates in the patients with mild-to-moderate and severe disease progression.

Parameter	Disease Severity	*n*	Mean or Median	Standard Deviation or (Min–Max)	*p* Value
Liver size (mm)	Mild-to-moderate	38	147.476	13.2861	>0.05
	Severe	10	146.610	20.7730	
Splenic volume (cm^3^)	Mild-to-moderate	38	288.25	105.00	>0.05
	Severe	10	319.25	117.59	
PV diameter (mm)	Mild-to-moderate	38	12.366	1.6355	>0.05
	Severe	10	12.350	2.2067	
SV diameter (mm)	Mild-to-moderate	38	6.600	(3.5–10.9)	>0.05
	Severe	10	6.900	(3.0–10.8)	
PV velocity (cm/s)	Mild-to-moderate	38	28.237	7.5837	>0.05
	Severe	10	26.710	7.2741	

**Table 4 medicina-55-00146-t004:** Liver size, spleen volume, portal vein and splenic vein diameters, and portal vein flow rates examined in 17 patients one month after discharge.

Parameter	Discharge	Mean	Standard Deviation	*p* Value
Liver size (mm)	Hospitalized	147.28	15.28	0.041
	Follow-up visit	137.65	15.60	
Splenic volume (cm^3^)	Hospitalized	281.40	97.84	0.013
	Follow-up visit	214.47	121.88	
PV diameter (mm)	Hospitalized	12.68	1.76	<0.001
	Follow-up visit	9.50	1.43	
SV diameter (mm)	Hospitalized	6.97	1.90	<0.001
	Follow-up visit	4.68	1.97	
PV velocity (cm/s)	Hospitalized	27.58	10.08	0.058
	Follow-up visit	22.18	5.22	

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
