# Peer review of "Evaluation of Portal Vein Doppler Findings in Patients with Crimean–Congo Hemorrhagic Fever"

_medicina, 2019, doi:10.3390/medicina55050146_

Round 1

Reviewer 1 Report

General comments:

In “Evaluation of Portal Vein Doppler Findings in Patients with Crimean-Congo Hemorrhagic Fever” KaravaĹź and Karakeçili describe the use of portal vein Doppler ultrasound to investigate its practicability in prediction of CCHF disease severity. Study was performed on a cohort of 99 patients (48 with disease and 51 healthy controls) over a near 2-year period. While no significant differences were observed in variables between mild/moderate and severe CCHF disease, differences were described from the healthy controls. The paper is informative, although the data is primarily descriptive with minimal effort taken to both determine the specifics of the findings beyond describing the observed differences, and to discuss the consequences of these in terms of CCHF disease. The paper is well written and organized, although there are sections that would benefit from additional editing and information.

General Comments:

In the methods section it should be stated how the laboratory findings (CBC, chemistry, INR etc) were obtained: diagnostic platforms, assays, manufacturers.

The results section first paragraph is a description of the contents of Table 1. While this information is very useful to the field in listing common and uncommon CCHFV presentations, the text adds very little to what can easily be obtained from the table. 

I would recommend modifying Table 1 with an additional set of columns in which the data for the 10 severe patients and 38 mild-to-moderate patients was separated, increasing the usefulness of this table.

The discussion comments of why two US findings were not reported more frequently in your cohort, and concluded it was due to earlier US than other studies. What day post symptom onset was the patient in this study with reported abdominal free fluid, and the thicken gallbladder wall? Were the US performed in these patients later than the others? These data should be included in this section.

Minor comments:

Figure 1: y-axes need units, and legend needs to be more descriptive to explain plots and dots.

Line 31: Update CCHFV nomenclature - family Nairoviridae, genus Orthonairovirus.

Line 54-56: Clarify sentence as wording is a little unclear.

Author Response

Response to Reviewer 1 Comments

General Comments:

Point 1: In the methods section it should be stated how the laboratory findings (CBC, chemistry, INR etc) were obtained: diagnostic platforms, assays, manufacturers.

Response 1: We made suggested corrections as requested by the reviewer (line 62-66).

Laboratory tests were closely monitored at the time of admission and during daily follow-up examinations. Accordingly, haemoglobin, platelet count, white blood cells (Sysmex XN 2000, Toa Medical Electronics Ltd, 2016, Japan), aspartate aminotransferase, alanine aminotransferase, creatinine phosphokinase, lactate dehydrogenase and international normalized rate (INR) (Beckman Coulter AU 2700, Beckman AU, 2014, Ireland) were recorded.

Point 2: The results section first paragraph is a description of the contents of Table 1. While this information is very useful to the field in listing common and uncommon CCHFV presentations, the text adds very little to what can easily be obtained from the table.

Response 2:  We made suggested corrections as requested by the reviewer and data on mental fog, hematuria gastrointestinal system hemorrhage were removed from the text (Line 106).

Point 3: I would recommend modifying Table 1 with an additional set of columns in which the data for the 10 severe patients and 38 mild-to-moderate patients was separated, increasing the usefulness of this table.

Response 3: We made suggested corrections as requested by the reviewer. Figure 1 (Line 117).

Point 4: The discussion comments of why two US findings were not reported more frequently in your cohort, and concluded it was due to earlier US than other studies. What day post symptom onset was the patient in this study with reported abdominal free fluid, and the thicken gallbladder wall? Were the US performed in these patients later than the others? These data should be included in this section.

Response 4: There are limited studies in the literature on radiological findings of patients with CCHF. The statistics on the increase thickened gallbladder wall and free fluid and show a significant difference with other studies. We needed to investigate this question. Only one study from this limited number of studies was given the time to develop symptoms and the first radiological evaluation. We think that comparison with data in a single study is an important limitation.

One patient had abdominal free fluid and US was performed five days after symptom onset. Because the single case was not statistically significant, we did not find the discussion based on this patient's data.

Minor comments:

Point 1: Figure 1: y-axes need units, and legend needs to be more descriptive to explain plots and dots.

Response 1: We made suggested corrections as requested by the reviewer. Figure 1 (Line 117).

Point 2: Line 31: Update CCHFV nomenclature - family Nairoviridae, genus Orthonairovirus.

Response 2: We made suggested corrections as requested by the reviewer (Line 31-32).

“It is caused by the CCHF virus, which belongs to the Orthonairovirus genus of the Nairoviridae family.”

Point 3: Line 54-56: Clarify sentence as wording is a little unclear.

Response 3: We made suggested corrections as requested by the reviewer (Line 54-56).

“Only seventeen of the recovered patients came to follow-up visit and Doppler US examinations were performed.”

Reviewer 2 Report

The aim of this study is not fully explained. Is there an expected specific feature belong to CCHF disease in portal vein dopler findings? How is the change in portal vein diameter affecting the course of the disease?  What is it the role in pathogenesis? What is benefit for the patient ?  Is there a portal vein doppler measurement  in other viral hemorrhagic fever?

The purpose, outputs and discussion of the study should be rewritten.

Author Response

Response to Reviewer 2 Comments

Point 1: The aim of this study is not fully explained. What is benefit for the patient ? 

Response 1: CCHF remains a serious seasonal problem in the study area and in certain endemic regions of Turkey. A worldwide recognized vaccine with proven efficacy to protect against the disease, or a specific antiviral drug that can be used in treatment, has not yet been developed. Recognition of the disease at an early stage and, especially a rapid initiation of supportive treatment, constitute the most important steps of treatment. In addition to supportive therapy, fresh frozen plasma is recommended for patients with bleeding, aPTT>60 sec, and INR>1.5. Also, patients with platelet counts <20.000/mm3 should be given thrombocyte suspension, patients with anemia and, if necessary, an erythrocyte suspension [1–2]

1.         Karakecili, F.; Cikman, A.; Aydin, M.; Binay, U.; Kesik, O.; Ozcicek, F. Evaluation of Epidemiological, Clinical, and Laboratory Characteristics and Mortality Rate of Patients with Crimean-Congo Hemorrhagic Fever in the Northeast Region of Turkey. J. Vector Borne Dis. 2018, 55 (3), 215. https://doi.org/10.4103/0972-9062.249479.

2.         Ergonul, O. Crimean-Congo Haemorrhagic Fever: Treatment and Use of Ribavirin. Klimik Dergisi/Klimik J. 2016, 29 (1), 2–9. https://doi.org/10.5152/kd.2016.02.

Point 2: Is there an expected specific feature belong to CCHF disease in portal vein dopler findings? How is the change in portal vein diameter affecting the course of the disease. What is it the role in pathogenesis?

Response 2: We are asking the same question with the reviewer and this is one of the important questions we are looking for in our study. A limited number of radiological studies are available on CCHF. It also causes heptomegaly and splenomegaly affecting the pathogen hematopoietic system. Therefore, we aimed to reveal the situation with radiological imaging by thinking that the portal system will be affected.

Point 3: Is there a portal vein doppler measurement in other viral hemorrhagic fever?

Response 3: Although there are a limited number of study evaluating ultrasound findings in other diseases of viral hemorrhagic fever, we could not find any publication except the case report evaluating portal vein doppler findings[1]. In our study, data such as intraabdominal free fluid, thickening of the gallbladder wall which is common radiological finding of viral haemorrhagic fever diseases and CCHF are rarely seen. However, there is not enough publication about the evaluation of ultrasound and Doppler findings related to viral hemorrhagic fever diseases and the fact that CCHF patients cannot be compared with other Viral Hemorrhagic fever diseases is an important limitation for our study.

1.        Thulkar, S.; Sharma, S.; Srivastava, D. N.; Sharma, S. K.; Berry, M.; Pandey, R. M. Sonographic Findings in Grade III Dengue Hemorrhagic Fever in Adults. J. Clin. Ultrasound 2000, 28 (1), 34–37. https://doi.org/10.1002/(SICI)1097-0096(200001)28:1<34::AID-JCU5>3.0.CO;2-D.

Point 4: The purpose, outputs and discussion of the study should be rewritten.

Response 4: Conclusions were revised (line 214-218).

Reviewer 3 Report

It is an original paper that aims to incorporate the ultrasounds findings (portal vein doppler) building a score as predictor of evolution severity criterion in patients with Crimean-Congo hemorrhagic fever. The authors think that the findings can contribute to prediction of the clinical course of CCHF in early period.

I believe that it could be interesting to describe the findings of the doppler study in different stages of the disease.

Author Response

Response to Reviewer 3 Comments

Comments and Suggestions for Authors:

It is an original paper that aims to incorporate the ultrasounds findings (portal vein doppler) building a score as predictor of evolution severity criterion in patients with Crimean-Congo hemorrhagic fever. The authors think that the findings can contribute to prediction of the clinical course of CCHF in early period.

I believe that it could be interesting to describe the findings of the doppler study in different stages of the disease.

Response: Thank you for your positive feedback.

Yours sincerely

Round 2

Reviewer 2 Report

No Comment

Author Response

Response: Thank you for your positive feedback.

Yours sincerely.